# Role of Liver Kinase 1B in Platelet Activation and Host Defense During *Klebsiella pneumoniae*-Induced Pneumosepsis

**DOI:** 10.3390/ijms26083714

**Published:** 2025-04-14

**Authors:** Osoul Chouchane, Valentine Léopold, Christine C. A. van Linge, Alex F. de Vos, Joris J. T. H. Roelofs, Cornelis van ‘t Veer, Tom van der Poll

**Affiliations:** 1Center of Infection and Molecular Medicine, Amsterdam University Medical Center, Location Academic Medical Center, University of Amsterdam, 1105 AZ Amsterdam, The Netherlands; o.chouchane@amsterdamumc.nl (O.C.); v.leopold@amsterdamumc.nl (V.L.); c.c.vanlinge@amsterdamumc.nl (C.C.A.v.L.); a.f.devos@amsterdamumc.nl (A.F.d.V.); c.vantveer@amsterdamumc.nl (C.v.‘t.V.); 2Department of Pathology, Amsterdam University Medical Center, Location Academic Medical Center, University of Amsterdam, 1105 AZ Amsterdam, The Netherlands; j.j.roelofs@amsterdamumc.nl; 3Division of Infectious Diseases, Amsterdam University Medical Center, Location Academic Medical Center, University of Amsterdam, 1105 AZ Amsterdam, The Netherlands

**Keywords:** microbial infection, microbes–host interaction, molecular and cell biology

## Abstract

Pneumonia is the most common cause of sepsis, with *Klebsiella pneumoniae* frequently implicated as a causative pathogen. Platelets play a crucial role in host defense during sepsis, and their activation is essential for effective immune responses, which is at least in part induced through activation of the collagen receptor glycoprotein (GP)VI. Platelets require energy for their activation, and Liver kinase B1 (LKB1) is a key regulator of energy metabolism. We sought to determine the role of LKB1 in platelet function and host response during *K. pneumoniae*-induced pneumosepsis. Platelet-specific-Lkb1-deficient mice were generated and compared to control littermates. Platelet counts were unaffected by Lkb1 deficiency in naïve mice. However, Lkb1-deficient platelets exhibited significant hyperreactivity to GPVI stimulation, an effect not observed after stimulation of the thrombin receptor protease-activated receptor 4. During *K. pneumoniae* infection, platelets of both Lkb1-deficient and control mice became equally hyporesponsive to GPVI stimulation, without differences between genotypes. Platelet-specific Lkb1 deficiency did not alter bacterial outgrowth or dissemination, inflammatory responses, or lung pathology. These findings suggest that while Lkb1 plays a role in regulating platelet activation in response to GPVI stimulation, it does not significantly impact platelet activation or the host response during pneumonia-induced sepsis.

## 1. Introduction

Sepsis is a syndrome defined by a dysregulated host response to infection that presents with organ damage and high mortality rates [1]. The most common source of sepsis is pneumonia [2], and *Klebsiella (K.) pneumoniae* is a frequent Gram-negative pathogen causative of pneumonia-induced sepsis [3].

Platelets play a crucial role in the pathophysiology of sepsis, a role extending beyond their traditional hemostatic function [4]. During sepsis, platelets become activated due to the ongoing coagulation cascade, inflammatory response, and endothelial tissue damage. These activated platelets contribute to hyperinflammation, disseminated intravascular coagulation, and microthrombosis, with the sequence ultimately leading to multiple organ failure. Moreover, platelets can interact with innate immune cells, modulating their functions and promoting a pro-inflammatory phenotype [4]. Additionally, platelets release pro-inflammatory peptides and cytokines, further influencing the immune response. The complex interplay between platelets, coagulation, and inflammation in sepsis makes the first a potential target for therapeutic interventions [5].

Our group previously reported a strongly impaired host defense in mice with thrombocytopenia [6] or deficiencies for the platelet receptors glycoprotein (GP)VI [7] or P-selectin [8] in experimental sepsis provoked by *K. pneumoniae*-induced pneumonia. In agreement with this finding, it has been reported that thrombin-stimulated platelets enhanced the killing of *K. pneumoniae* by human monocytic cells in vitro [9].

The energy metabolism of platelets is fundamental to their activation [10]. Platelets are the most metabolically active ‘cells’ in the blood compartment under homeostatic conditions [11]. A recent study demonstrated that platelets have an increased energy demand during activation, and that mice with platelet-specific deletion of pyruvate kinase M2, a key facilitator of glycolysis, display diminished activation capabilities [12]. Conversely, in vitro studies have indicated that platelets exhibit metabolic flexibility; when deprived of one energy source, they can adapt by switching to alternative substrates to sustain their activation needs [10,13]. Yet, in vivo studies confirming this are lacking.

Liver kinase B1 (LKB1, or serine-threonine kinase 11 (STK11)), is a master kinase and the primary kinase that activates AMP-activated protein kinase (AMPK) [14,15]. AMPK is a crucial metabolic regulator within cells. Through the activation of AMPK, LKB1 steers cells into a catabolic mode conducive to ATP production in situations of low energy availability or increased energy demand [15,16]. In this study we generated platelet-specific-Lkb1-deficient mice to explore the role of platelet Lkb1 in platelet activation and the host response to pneumonia caused by *K. pneumoniae*.

## 2. Results

### 2.1. Lkb1 Deficiency in Platelets Is Associated with Platelet Hyperactivation in Response to GPVI Stimulation

To elucidate the role of Lkb1 in platelet function, we crossed *Stk11^fl/fl^* mice with *Pf4*-Cre mice to generate platelet-specific-Lkb1-deficient mice; *Stk11^fl/fl^* Cre-negative littermates were used as controls. Knock-down of Lkb1 in platelets was established by Western blot on platelet lysates (Figure 1A). Platelet-specific Lkb1 deficiency did not affect platelet counts in naïve mice (Figure 1B, see bars labelled “Naïve” on the left side of this panel). We then evaluated the impact of Lkb1 deficiency on platelet activation in naïve mice in response to platelet agonists stimulating the collagen receptor GPVI (CRP-XL) or the thrombin receptor PAR4 (PAR4AP) by measuring the surface expression of several proteins: P-selectin (CD62P), stored in platelet α granules under resting conditions and crucial for platelet-leukocyte interactions, served as a marker of platelet activation and α degranulation; CD63, a transmembrane protein present in dense (δ) granules and released upon platelet activation, was utilized as a marker of dense granule release; and active GPIIbIIIa, employed by platelets for binding fibrinogen and von Willebrand factor, was used as another marker for platelet activation [17,18,19]. The surface expression of these markers did not differ between resting Lkb1-deficient and control platelets (Figure 1C, see left panels, labeled PBS-BSA, and bars labeled “Naïve”). The expression of P-selectin, CD63, and active GPIIbIIIa increased in response to either CRP-XL or PAR4AP on both Lkb1-deficient and control platelets (Figure 1C, see panels labeled with these respective agonists, and bars labeled “Naïve”). Lkb1-deficient platelets exhibited a significant hyperreactivity to GPVI stimulation at both high and low dosages of CRP-XL, as evidenced by increased surface P-selectin expression compared to control platelets. Similar trends were found for CRP-XL-induced CD63 surface expression and active GPIIbIIIa, albeit nonsignificant. In contrast, PAR4AP induced expression of activation markers did not differ between Lkb1-deficient and control platelets. Complex formation between platelets and monocytes did not differ in whole blood obtained from platelet-specific-Lkb1-deficient and littermate control mice (Appendix A, left panels, labeled “Naïve”).

### 2.2. Platelet-Specific Lkb1 Deficiency Does Not Impact the Host Response During Pneumosepsis

Our group previously reported strong constitutive expression of the GPVI ligand collagen in the lung, together with induction of the GPVI ligands fibrin and histones after infection with the common respiratory pathogen *K. pneumoniae* via the airways [7]. Importantly, we documented a clear protective role of platelet GPVI during pneumosepsis induced by *K. pneumoniae*, as reflected by increased bacterial growth and dissemination, and reduced platelet activation in GPVI-depleted or GPVI-deficient mice [7]. In light of the role of Lkb1 in the GPVI responsiveness of resting platelets, we considered it of interest to study the impact of platelet-specific Lkb1 deficiency on the host response during *Klebsiella*-induced pneumosepsis, using the same model as in our earlier study [7]. *Stk11^fl/fl^* × *Pf4*-Cre and control mice were infected with *K. pneumoniae* via the airways, and sacrificed 24 or 40 h after infection. Infection resulted in a significant drop in platelet counts, which, however, was not different between the two groups (Figure 1B). At 40 h after infection, platelet CD63 expression was reduced relative to pre-infection, yet not different between *Stk11^fl/fl^* × *Pf4*-Cre and control mice (Figure 1C). At this post-infection timepoint, the expression levels of P-selectin and GPIIbIIIa were not different from those in naïve mice, and similar in Stk11fl/fl × Pf4-Cre and control mice (Figure 1C). Moreover, at this timepoint, there were no differences between genotypes regarding responsiveness to CRP-XL or PAR4AP (Figure 1C), indicating that the hyperreactive phenotype of Lkb1-deficient platelets after GPVI stimulation was not maintained after infection. Notably, platelet responsiveness to GPVI stimulation in particular was strongly reduced in infected mice, irrespective of genotype (*p* < 0.001 for both genotypes for CRP-XL 1.0 μg/mL and CRP-XL 0.05 μg/mL). In contrast, platelet responsiveness to stimulation of the PAR4 thrombin receptor did not decrease after infection; platelet CD63 expression was even enhanced at 40 h after infection, relative to baseline, and again irrespective of genotype (*p* < 0.01 for both genotypes for PAR4AP 0.1 mg/mL and *p* < 0.001 for PAR4AP 0.2 mg/mL). To determine the temporal dynamics of the role of Lkb1 in platelets’ responsiveness to GPVI stimulation during *K. pneumoniae*-induced pneumosepsis, we evaluated platelet activation at 24 h post-infection (Appendix A). At this earlier timepoint, we began to observe the diminishing of the hyperreactivity of Lkb1-deficient platelets to GPVI stimulation, particularly with regard to the expression of P-selectin (*p* = 0.0208 for the difference between genotypes at CRP-XL 1.0 µg/mL; *p* = 0.10 after correcting for multiple testing). Complex formation between platelets and monocytes did not differ between platelet-specific-Lkb1-deficient and littermate control mice at 40 h after infection (Appendix A, right panels).

Platelet-specific Lkb1 deficiency did not impact bacterial outgrowth at the primary site of infection (lung tissue and BALF; Figure 2A) or distant organs (Figure 2B). Likewise, the concentrations of cytokines in BALF or plasma, neutrophil influx into BALF (Table 1) and severity of lung pathology (Figure 3; see Appendix A for individual components of the pathology score) did not differ between *Stk11^fl/fl^* × *Pf4*-Cre and control mice.

## 3. Discussion

In this study, by generating platelet-specific-Lkb1-deficient mice, we document an inhibitory role of Lkb1 in platelet activation, as indicated by P-selectin expression, upon stimulation of the collagen receptor GPVI in vitro. Considering the earlier studies from our group that established the protective roles of platelet GPVI [7] and P-selectin [8] in the host response during *K. pneumoniae*-induced pneumosepsis, this finding prompted us to investigate the involvement of platelet Lkb1 in this model. Yet, our data indicate that platelet-specific Lkb1 deficiency does not affect platelet responses or host defense during *K. pneumoniae* infection, suggesting that alternative pathways or compensatory mechanisms may supersede or mask Lkb1’s function in the complex physiological context of a live organism facing a bacterial challenge. In this context, it should be noted that platelets can be activated through a variety of agonists and mechanisms during sepsis, including adenosine diphosphate, thromboxane A2, cytokines, matrix metalloproteinases, acidosis, and interactions with endothelial cells, conditions that are not captured or are incompletely captured in ‘ex vivo’ stimulations [4].

One possible explanation for the hyperactivation in GPVI-stimulated Lkb1-deficient platelets may stem from studies on mast cells, in which AMPK deficiency—AMPK being a key target of Lkb1—leads to heightened FcεRI-mediated activation [20]. In mast cells, the absence of AMPK regulation enhances Syk kinase signaling, thereby increasing activation levels [20]. The signaling cascade initiated by GPVI stimulation in platelets mirrors the FcεRI-mediated activation in mast cells, as it involves phosphorylation of FcRγ ITAM, recruitment and activation of Syk kinase, and downstream activation of phospholipase C [21,22,23]. Similarly, the hyperreactivity in Lkb1-deficient platelets could result from the lack of AMPK-mediated inhibition of Syk kinase. Notably, upon infection with *K. pneumoniae*, the hyperreactive phenotype of Lkb1-deficient platelets disappeared, and instead, a strong hyporeactive phenotype was observed upon CRP-XL stimulation in both groups. This diminishing effect of infection on platelet reactivity through GPVI has been previously noted by our group [24]. Further research is warranted to reveal the molecular mechanisms by which platelet GPVI responsiveness is regulated by Lkb1 during bacterial infection.

Platelets can interact with monocytes, a process which can modify monocyte functions [25]. An earlier study reported that—while platelets did not directly inhibit the growth of *K. pneumoniae*—thrombin-stimulated human whole blood inhibited the growth of *K. pneumoniae* in a platelet-dependent manner [9]. Thrombin increased complex formation between platelets and monocytes in whole blood, and thrombin-stimulated platelets enhanced killing of *K. pneumoniae* by U937 monocytic cells [9]. In this study, we show that platelet-specific Lkb1 deficiency does not impact the capacity of platelets to form complexes with monocytes in either naïve mice or mice infected with *K. pneumoniae*.

The previous research has reported that thrombin stimulation of platelets leads to increased phosphorylation of Lkb1, which in turn activates AMPK [26]. AMPK activation was shown to contribute to platelet responsiveness, as pharmacologic inhibition of this kinase attenuated thrombin-induced platelet aggregation and clot retraction. In agreement, platelets from AMPKα2-deficient mice showed impaired clot retraction; moreover, rebleeding was more frequent and FeCl_3_-induced thrombi less stable in AMPKα2-deficient mice. AMPKα2 likely impacts platelet function and thrombus stabilization by regulating the Fyn-mediated phosphorylation of β3 integrin (i.e., the GPIIIa subunit of GPIIbIIIa) [26]. We did not detect a role for Lkb1 in thrombin-induced expression of P-selectin, CD63, or active GPIIbIIIa at the surface of the platelets. While this does not exclude a role for Lkb1 in other platelet responses, our results do show that in severe infection, such as induced by *K. pneumoniae*, platelet Lkb1 is redundant for induction of inflammation and host defense mechanisms.

Our study has limitations. While differences between genotypes after infection were not significant, some trends were seen (for example, higher median CFU counts in lungs and BALF, and higher median plasma IL-6 levels in platelet-specific-Lkb1-deficient mice at 40 h); expansion of the sample size would possibly reveal significant differences between groups. We did not determine the impact of platelet Lkb1 on survival, due to ethical restrictions in our country.

Previous work has suggested that therapeutic interventions that target activated platelets might reduce hyperinflammatory and procoagulant aberrations in sepsis [4,5]. We here report experimental data that argue against a significant role for Lkb1 in platelet activation and host defense in pneumonia-derived sepsis.

## 4. Materials and Methods

### 4.1. Animals

We generated platelet-specific (and megakaryocyte-specific) Lkb1-deficient mice by crossing mice homozygous for the *Stk11* flox allele (*Stk11^fl/fl^*) (014143; Jackson Laboratory, Bar Harbor, ME) with mice expressing Cre recombinase under the control of the platelet factor 4 promoter (*Pf4*-Cre mice, 008535; Jackson Laboratory). Control mice utilized in all experiments were *Stk11^fl/fl^* × *Pf4*-Cre-negative littermates. All mice underwent a minimum of 6 backcrosses to a C57Bl/6 background and were maintained under standard care conditions. In this article, the platelet-specific-Lkb1-deficient mice are referred to as such or as *Stk11^fl/fl^* × *Pf4*-Cre mice; Cre-negative littermates are referred to as control or *Stk11^fl/fl^* mice. Experimental cohorts consisted of age- and sex-matched mice aged 8–12 weeks. Mice were assigned random numbers during experiments, but group allocation was revealed during data analyses. All experimental procedures were conducted in compliance with the Dutch Experiment on Animals Act and received approval from the Central Commission for Animal Experiments.

### 4.2. Platelet Isolation and Preparation of Washed Platelets

Blood was collected from the inferior vena cava with a 21-gauge needle and diluted 1:5 with citrate. To obtain platelet-rich plasma (PRP), blood was centrifuged at room temperature at 180× *g* for 15 min, after which a 1:5 volume of acid citrate dextrose was added to prevent platelet activation. Washed platelets were prepared by spinning platelet-rich plasma at 800 g for 10 min, resuspending the pellet in buffer A (137 mM NaCl, 12 mM NaHCO_3_, 2.6 mM KCL, 1 mM MgCl_2_-5H_20_, 5 mM glucose and 2 mM EDTA, pH 6.5), and counting using Precision Count Beads (Biolegend, San Diego, CA, USA) and a Cytoflex S flow cytometer (Beckman Coulter, Brea, CA, USA). Platelets were subsequently washed and resuspended at 400 × 10^6^ platelets/mL (a concentration representative of physiological platelet levels) in buffer B (137 mM NaCl, 12 mM NaHCO_3_, 2.6 mM KCL, 1 mM MgCl_2_-5H_20_, 5 mM glucose, pH 7.35). Washed platelets were lysed with SDS-PAGE sample buffer and stored at −20 °C until analysis.

### 4.3. Western Blotting

Western blotting was performed following a standard protocol as previously described [27], using rabbit anti-LKB1 (D60C5) and rabbit anti-β-actin (4967L) (CellSignaling, Danvers, MA, USA). Goat anti-rabbit antibody (7074S; Cell Signaling) conjugated with horseradish peroxidase was used as secondary antibody.

### 4.4. Flow Cytometry

Platelet activation was assessed through flow cytometry. For this, citrated whole blood was incubated with phosphate-buffered saline (PBS) with bovine serum albumin (BSA) 0.1% (unstimulated), the GPVI agonist cross-linked collagen-related peptide (CRP-XL, CambCol-Laboratory, Cambridge, United Kingdom), or the protease-activated receptor 4 agonist (PAR4AP, Bachem, Bubendorf, Switzerland) for 30 min at room temperature. Simultaneously, platelets were stained with anti-CD61-APC (clone 2C9.G2, Biolegend), anti-CD62P-FITC (BD Biosciences, San Jose, CA, USA), anti-CD63-PerCP-Cy5.5 (Clone: NVG-2, Biolegend), and anti-JON/A-PE (detecting active GPIIbIIIa) (Emfret Analytics, Eibelstadt, Germany). Activation was terminated by the addition of 8 volumes of PBS containing 0.3% paraformaldehyde. Due to the large number of events in whole blood, we diluted the stained samples 1:4 in PBS and proceeded to immediate analysis using flow cytometry on a CytoFLEX S (Beckman Coulter) until 10,000 CD61+ events were recorded. At the 24 h timepoint, flow cytometry analysis was performed on a FACSCanto II (BD Biosciences). We used Precision Count Beads™ (Biolegend) to assess platelet counts in whole blood. Platelet–monocyte complexes were analyzed in PBS-diluted whole blood using the following antibodies: anti-CD4-FITC, anti-Ly6C-AF700, anti-Ly6G-APC, and anti-CD11B-PECy7 (BD Biosciences). The blood was diluted 1:15 in PBS containing the antibody mix and incubated for 30 min at room temperature. After incubation, the sample was centrifuged at 1000× *g* for 2 min, fixed, and lysed at 4 °C with Lyse/Fix Buffer (BD Biosciences). The mixture was then spun again, resuspended in PBS, and analyzed by flow cytometry; platelet–monocyte complexes were quantified by the MFI of CD41 on gated monocytes. Neutrophil influx into bronchoalveolar lavage fluid (BALF) was quantified by staining with anti-CD11b-PECy7 (BD Biosciences) and anti-Ly6G-PE (eBiosiences, San Diego, CA, USA) for 30 min. Data analysis was performed using FlowJoTM v10 (BD Biosciences). We assessed compensation settings using single-stained controls. We assessed the percentage of positivity for a surface marker of interest using fluorescence-minus-one controls.

### 4.5. Mouse Infection Model

Pneumonia and sepsis were induced by intranasal inoculation with *K. pneumoniae* serotype 2 (ATCC 43816, Rockville, MD, USA; 10^4^ colony-forming units [CFUs]). Infection and processing of organs were performed as described elsewhere [6,7,8]. In short, mice were euthanized at 24 or 40 h following *K. pneumoniae* infection for the collection of blood, lungs, spleen, and liver. We determined bacterial loads by counting CFU from serial dilutions plated on blood agar plates incubated at 37 °C for 16 h. In mice euthanized at 40 h following *K. pneumoniae* infection, we also collected bronchoalveolar lavage fluid (BALF) prior to collection of the lungs.

### 4.6. Assays

In plasma and BALF, interleukin (IL)-6, tumor necrosis factor (TNF), IL-10, and chemokine (C-C motif) ligand 2 (CCL2) were measured using a Cytometric Bead Array (Mouse Inflammation Kit; BD Biosciences, Franklin Lakes, NJ, USA) on a CytoFLEX S (Beckman Coulter, Brea, CA, USA).

### 4.7. Pathology

The left lung lobe was fixed in 10% buffered formaldehyde and embedded in paraffin, after which, sections 4 μm thick were cut. Slides were stained with hematoxylin and eosin (H/E). Slides were anonymized and scored by a pathologist blinded as to group identity. In lung tissue the following parameters were scored on a scale from 0 (absent) to 4 (very severe): presence of thrombi, interstitial inflammation, endothelitis, bronchitis, oedema, pleuritis, and bleeding. The pathology score is the sum of all scores for each organ, respectively.

### 4.8. Ethical Statement

Experiments were performed in accordance with the Experiments on Animals Act of the Netherlands and relevant European Directives, and approved by the Central Authority for Animal Experiments and the Animal Welfare Body of the Amsterdam University Medical Center (approval numbers 174125-186 and -1100, approval date of project: 16 January 2017; approval number 2216434-106, approval date of project: 25 November 2022). The study was conducted in accordance with local legislation and institutional requirements.

### 4.9. Statistics

Group comparisons were carried out using the Mann–Whitney U test. The *p*-values were adjusted for multiple testing using the Benjamini–Hochberg method, with a significance threshold set at *p* < 0.05. Statistical analyses were conducted using Rstudio version 4.4.

## Figures and Tables

**Figure 1 ijms-26-03714-f001:**
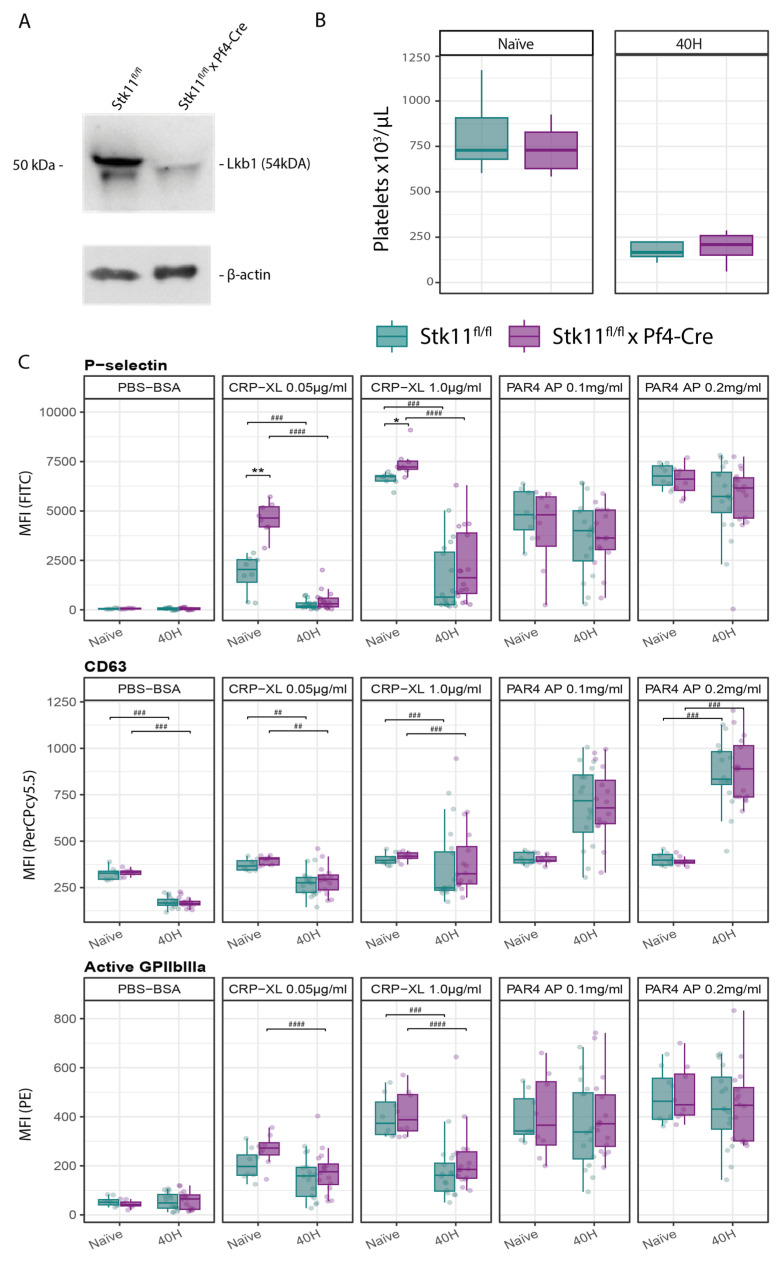
Generation of platelet-specific-Lkb1-deficient mice and the effect of platelet-specific Lkb1 deficiency on platelet counts and activation in naïve mice and mice with *Klebsiella*-induced pneumosepsis. (**A**) Representative Western blot of platelet lysates for Lkb1, showing effective knockdown of platelet Lkb1 in *Stk11^fl/fl^* × *Pf4-Cre* mice. (**B**) Boxplots displaying the platelet counts in blood obtained in naïve (uninfected) mice and mice 40 h after infection with *Klebsiella pneumoniae* via the airways; *n* = 8 per genotype for each timepoint. (**C**) Boxplots comparing the median fluorescence intensity (MFI) for P-selectin, CD63, or GPIIbIIIa in active conformation on the platelet surface of *Stk11^fl/fl^* × *Pf4-Cre* and *Stk11^fl/fl^* littermate control mice (naïve and 40 h after infection). Each column represents a treatment with the platelet agonist indicated or vehicle (PBS-BSA); *n* = 8 per group for naïve mice and *n* = 16 per group for mice 40 h after infection. The *p*-values are derived from BH-adjusted Wilcoxon tests; *p*-values between genotypes at specific timepoints are denoted by ‘*’; * *p* < 0.05, ** *p* < 0.01. The *p*-values between timepoints within the same genotype are denoted by ‘#’; ## *p* < 0.01, ### *p* ≤ 0.001, #### *p* ≤ 0.0001.

**Figure 2 ijms-26-03714-f002:**
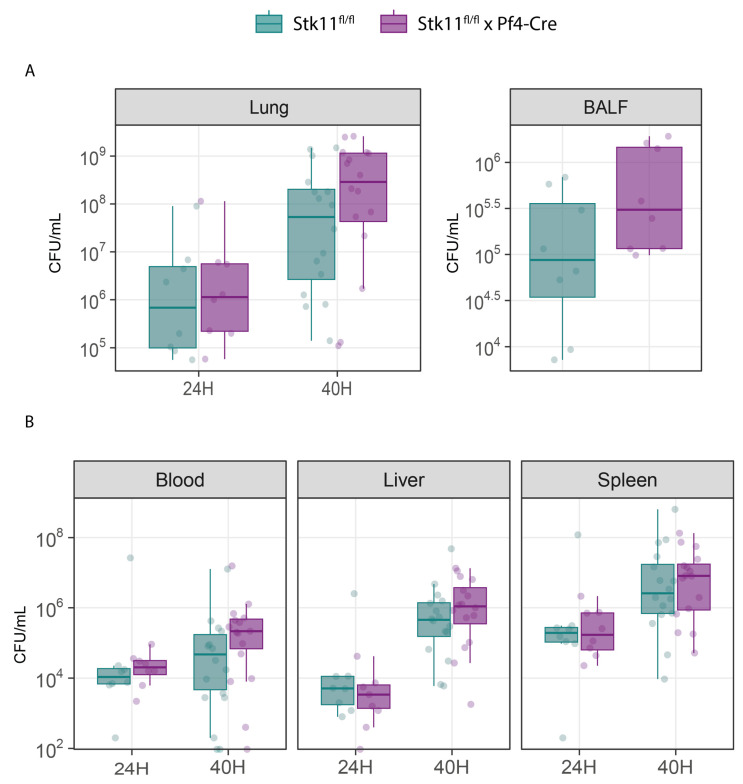
Platelet-specific Lkb1 deficiency does not impact bacterial growth or dissemination during *Klebsiella*-induced pneumosepsis. Bacterial counts are quantified as colony forming units (CFU) per ml at the primary site of infection and distant organs. (**A**) Lung and BALF and (**B**) Blood, Liver, and Spleen at 24 and 40 h after *K. pneumoniae* infection via the airways. (For (**B**), bacterial dissemination to blood, liver, and spleen is quantified as CFU per ml at 24 h and 40 h post *K. pneumonia* infection.) *n* = 8 per group at t = 24 h; *n* = 16 per group at t = 40 h.

**Figure 3 ijms-26-03714-f003:**
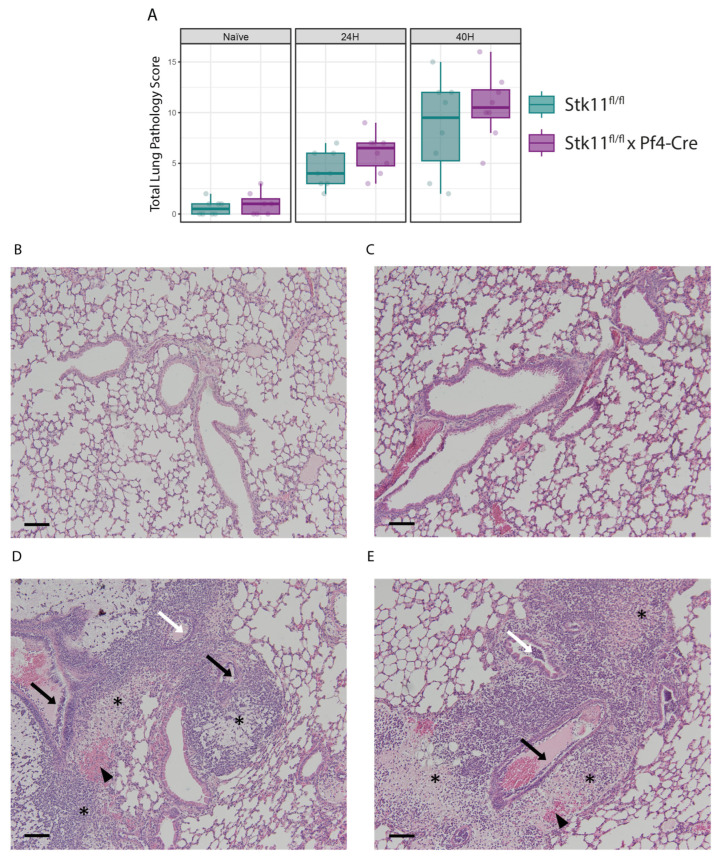
Pathology scores. (**A**) Lung pathology scores from H/E-stained slides, evaluated by a pathologist blinded to group identity, and visualized in three panels: non-infected naïve mice, and mice 24 and 40 h after *K. pneumoniae* infection. (**B**,**C**) Representative microphotographs (magnification ×20) of H/E-stained slides from the lungs of naïve mice, *Stk11^fl/fl^* mice, and *Stk11^fl/fl^* × *Pf4-Cre* mice, respectively. Scale bars represent 200 μm. (**D**,**E**) Representative microphotographs (magnification ×20) of H/E-stained slides of lungs of mice 40 h post *K. pneumoniae* infection via the airways; *Stk11^fl/fl^* and *Stk11^fl/fl^* × *Pf4-Cre* mice, respectively. Scale bars represent 200 μm. Asterisks represent perivascular and peribrochial edemas and collections of neutrophils in the peribronchovascular interstitium. Arrowheads point to interstitial hemorrhages. Black arrows indicate endothelitis. White arrows indicate bronchitis.

**Table 1 ijms-26-03714-t001:** Cytokine concentrations and neutrophil influx.

	*Stk11^fl^*^/*fl*^ Mice	*Stk11^fl/fl^* × *Pf4*-Cre Mice	*p*-Value
Plasma			
Naïve	*n* = *8*	*n* = *8*	
Interferon-γ (pg/mL)	12.6 [0.15]	12.3 [0.14]	0.16
IL-6 (pg/mL)	10.9 [1.13]	11.0 [2.91]	0.84
IL-10 (pg/mL)	8.8 [1.22]	8.83 [0.72]	0.80
CCL2 (pg/mL)	57.9 [0.97]	55.7 [5.67]	0.63
TNF (pg/mL)	24.9 [0.57]	23.9 [1.62]	0.47
24 h after *K. pneumoniae* infection	*n* = *8 **	*n* = *8*	
Interferon-γ (pg/mL)	14.5 [3.57]	13.7 [1.69]	0.66
IL-6 (pg/mL)	105.1 [140.52]	594.8 [439.12]	0.18
IL-10 (pg/mL)	13.6 [6.27]	26.1 [14.52]	0.16
CCL2 (pg/mL)	1479.3 [3568.42]	3008.1 [1576.81]	0.59
TNF (pg/mL)	68.7 [107.57]	137.4 [53.25]	0.50
40 h after *K. pneumoniae* infection	*n* = *16 **	*n* = *16 **	
Interferon-γ (pg/mL)	13.1 [4.68]	13.0 [3.32]	0.73
IL-6 (pg/mL)	289.0 [713.17]	374.6 [2192.03]	0.59
IL-10 (pg/mL)	20.1 [20.04]	30.0 [30.42]	0.59
CCL2 (pg/mL)	1105.9 [1139.02]	1137.3 [4404.23]	0.67
TNF (pg/mL)	62.2 [33.34]	75.3 [89.97]	0.90
Bronchoalveolar lavage fluid	*n* = *8*	*n* = *8*	
40 h after *K. pneumoniae* infection			
Interferon gamma (pg/mL)	12.4 [0.35]	12.5 [0.35]	0.68
IL-6 (pg/mL)	62.3 [143.39]	122.6 [176.15]	0.56
IL-10 (pg/mL)	11.8 [2.66]	12.6 [2.63]	0.56
CCL2 (pg/mL)	55.9 [24.05]	55.7 [19.18]	0.96
TNF (pg/mL)	34.5 [48.73]	119.8 [85.75]	0.50
Neutrophil counts (10^3^ cells/mL)	39.7 [44.2]	116.4 [126.9]	0.59

Data are presented as median [IQR]. The number of mice per group per experiment is shown in the grey bar. * Exceptions (due to technical issues): for plasma CCL2 at 40 h, *n* = 15 for *Stk11^fl/fl^* mice and *n* = 13 for *Stk11^fl/fl^* × *Pf4-Cre* mice; for plasma IL-6 at 40 h, *n* = 15 for both groups; for plasma IL-6, *n* = 7 for *Stk11^fl/fl^*; and for plasma CCL2 at 24 h, *n* = 6 and *n* = 7 for the two groups, respectively.

## Data Availability

Raw data are available upon request to the corresponding author.

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
