# Peer review of "Role of Liver Kinase 1B in Platelet Activation and Host Defense During Klebsiella pneumoniae-Induced Pneumosepsis"

_ijms, 2025, doi:10.3390/ijms26083714_

Round 1
Reviewer 1 Report
Comments and Suggestions for Authors
Authors investigated the role of LKB1 in platelet activation and host response in pneumonia model. Platelet is very important in sepsis pathophysiology, but there should be more mechanistic study. In that background, this study might be important. However, there are some issues to discuss.
They showed Lkb1 deficiency in platelets is associated with platelet hyperactivation in response to GPVI stimulation, even though I can’t see the figure.
However, they commented that Platelet specific Lkb1 deficiency does not impact the host response during pneumonia sepsis. However, in table 1, even though there is no statistical difference, there are some trends btw two groups, e.g. IL-6 (105 vs. 594). The same trends were observed in IL-10, CCL2, and TNF.
I am not sure two groups have no real difference or numbers are short.
In clinical aspects, what is the implication of the discrepancies btw platelet hyperactivation in response to GPVI stimulation and no difference in in vivo model?
Author Response
Reviewer 2
Authors investigated the role of LKB1 in platelet activation and host response in pneumonia model. Platelet is very important in sepsis pathophysiology, but there should be more mechanistic study. In that background, this study might be important. However, there are some issues to discuss.
Comment 1: They showed Lkb1 deficiency in platelets is associated with platelet hyperactivation in response to GPVI stimulation, even though I can’t see the figure. However, they commented that Platelet specific Lkb1 deficiency does not impact the host response during pneumonia sepsis. However, in table 1, even though there is no statistical difference, there are some trends btw two groups, e.g. IL-6 (105 vs. 594). The same trends were observed in IL-10, CCL2, and TNF. I am not sure two groups have no real difference or numbers are short.
Response 1: We now discuss this issue in a limitations paragraph in the Discussion:
Lines 225-230: “Our study has limitations. While differences between genotypes after infection were not significant, some trends were seen (for example higher median CFU counts in lungs and BALF, and higher median plasma IL-6 levels in platelet specific Lkb1-deficient mice at 40 hours); expansion of the sample size would possibly reveal significant differences between groups”.
Comment 2: In clinical aspects, what is the implication of the discrepancies btw platelet hyperactivation in response to GPVI stimulation and no difference in in vivo model?
Response 2: We addressed the discrepancy between our in vitro and in vivo data in the first paragraph of the Discussion: “Yet, our data indicate that platelet specific Lkb1 deficiency does not affect platelet responses or host defense during K. pneumoniae infection, suggesting that alternative pathways or compensatory mechanisms may supersede or mask Lkb1's function in the complex physiological context of a live organism facing a bacterial challenge”. We now add to this the following:
Lines 184-188: “In this context, it should be noted that platelets can be activated through a variety of agonists and mechanisms during sepsis, including adenosine diphosphate, thromboxane A2, cytokines, matrix metalloproteinases, acidosis and interactions with endothelial cells, conditions that are not or incompletely captured in “ex vivo” stimulations”.

Reviewer 2 Report
Comments and Suggestions for Authors
Findings presented in the paper suggest that while Lkb1 plays a role in regulating platelet activation in response to GPVI stimulation, it does not significantly impact platelet activation or the host response during pneumonia-induced sepsis.
In my opinion the paper is too concise. Even if the role of platelet in the infection can be interesting and extensevely analyzed, this is not the case. Moreover it is not clear what is the message that the authors want to give to the scientific community.
Here I resume my comments: The paper doi: 10.1128/iai.00556-22 described that activated platelets significantly enhanced monocyte-mediated killing of K. pneumoniae . I think that in this article the authros have to consider this mechanism and to include monocyte analysis. Figure 1 Representative dot plot for each marker analyzed need to be showed.Figure 2
The authors have to comment the results of CFU counts that are higher in spleen respect to blood. At 40 hours there is a tendency in the increase of CFU in Stk11fl/fl x Pf4-Cre mice in all the districts excpet spleen. Author have to comment also on this. It would be interesting to perform a survival curve.
Figure3
In the figures scale bar is missing and the maginification is not indicated nor in the legends nor in the methods section.
In the hematossilin eosin figures authors need to indicate with arrows or similar damage area.
Pathology score could be presented considering the single parameter indicated in the matherial and methods to see if all parameters are equal between the two groups or if someone of those is differentially represented even without a statistical significance. In this case the description is more informative of the damage: if the authors prefere they can present the single data in a supplementary figure.
In general the introduction is too short and recent reviews on the argument are missing.
In the discussion authors do not clarify snough how the results obtained can impact on the research regarding bacterial infection and sepsis.
Details regarding the role of platelets from the clinical point of view are missing.
Author Response
Reviewer 1
Findings presented in the paper suggest that while Lkb1 plays a role in regulating platelet activation in response to GPVI stimulation, it does not significantly impact platelet activation or the host response during pneumonia-induced sepsis. In my opinion the paper is too concise. Even if the role of platelet in the infection can be interesting and extensively analyzed, this is not the case. Moreover it is not clear what is the message that the authors want to give to the scientific community.
Comment 1: The paper doi: 10.1128/iai.00556-22 described that activated platelets significantly enhanced monocyte-mediated killing of K. pneumoniae . I think that in this article the authors have to consider this mechanism and to include monocyte analysis.
Response 1: We thank the reviewer for pointing our attention to this interesting and relevant paper. Main findings of this earlier study include that while platelets do not directly inhibit growth of K. pneumoniae, thrombin-stimulated whole blood inhibits growth of K. pneumoniae in a platelet-dependent manner. Additional studies showed that thrombin increased complex formation between platelets and monocytes in whole blood, and that thrombin-stimulated platelets enhanced killing of K. pneumoniae by U937 monocytic cells.
In response to this comment we now incorporated data not used in the original submission, showing that platelet-monocyte complex formation did not differ between platelet specific Lkb1 deficient mice and control mice at baseline or after infection. In addition, we now discuss the paper suggested by the reviewer in our manuscript. The following changes were made:
Introduction (lines 49-51): “In agreement, thrombin-stimulated platelets enhanced killing of K. pneumoniae by human monocytic cells in vitro” (reference as suggested added)”.
Results (lines 94-96): “Complex formation between platelets and monocytes did not differ in whole blood obtained from platelet specific Lkb1-deficient and littermate control mice (Supplementary Figure 1, left panels, labeled “naïve”)”.
Results (lines 141-143): “Complex formation between platelets and monocytes did not differ between platelet specific Lkb1-deficient and littermate control mice at 40 hours after infection (Supplementary Figure 1, right panels)”.
Discussion (lines 204-211): “Platelets can interact with monocytes, which can modify monocyte functions [new Ref. added]. An earlier study reported that - while platelets did not directly inhibit growth of K. pneumoniae - thrombin-stimulated human whole blood inhibited growth of K. pneumoniae in a platelet-dependent manner [Ref as suggested by the reviewer added]. Thrombin increased complex formation between platelets and monocytes in whole blood, and thrombin-stimulated platelets enhanced killing of K. pneumoniae by U937 monocytic cells. We here show that platelet specific Lkb1 deficiency does not impact the capacity of platelets to form complexes with monocytes in either naïve mice or mice infected with K. pneumoniae”.
Materials and Methods (lines 284-291): “Platelet-monocyte complexes were analyzed in PBS-diluted whole blood using the fol-lowing antibodies: anti-CD4-FITC, anti-Ly6C-AF700, anti-Ly6G-APC, and an-ti-CD11B-PECy7 (BD Biosciences). Blood was diluted 1:15 in PBS containing the antibody mix and incubated for 30 minutes at room temperature. After incubation, the sample was centrifuged at 1000g for 2 minutes, fixed, and lysed at 4°C with Lyse/Fix Buffer (BD Bio-sciences). The mixture was then spun again, resuspended in PBS, and analyzed by flow cytometry, platelet-monocyte complexed were quantified by the MFI of CD41 on gated monocytes.”
Comment 2: Figure 1 Representative dot plot for each marker analyzed need to be showed.
Response 2: This was done as requested. In addition, we now show individual data points (dots) in all applicable figures.
Comment 3: Figure 2. The authors have to comment the results of CFU counts that are higher in spleen respect to blood. At 40 hours there is a tendency in the increase of CFU in Stk11fl/fl x Pf4-Cre mice in all the districts except spleen. Author have to comment also on this. It would be interesting to perform a survival curve.
Response 3: CFU counts are typically higher in organs than in blood in this model (we reported this in all our earlier publications using it), probably because bacteria are captured in organs, not in the circulation. We prefer not to discuss this in the manuscript, since this finding is common to this model and does not relate to a role of platelet Lkb1. Differences in CFU’s between genotypes were all not even close to significant. We cannot perform survival studies in our country because of ethical restrictions. We now do address these comments in a limitations paragraph in the Discussion:
Lines…: “Our study has limitations. While differences between genotypes after infection were not significant, some trends were seen (for example higher median CFU counts in lungs and BALF, and higher median plasma IL-6 levels in platelet specific Lkb1-deficient mice at 40 hours); expansion of the sample size would possibly reveal significant differences between groups. We did not determine the impact of platelet Lkb1 on survival due to ethical restrictions in our country”.
Comment 4: Figure 3 - in the figures scale bar is missing and the magnification is not indicated nor in the legends nor in the methods section. In the hematoxylin eosin figures authors need to indicate with arrows or similar damage area. Pathology score could be presented considering the single parameter indicated in the materials and methods to see if all parameters are equal between the two groups or if someone of those is differentially represented even without a statistical significance. In this case the description is more informative of the damage: if the authors prefer they can present the single data in a supplementary figure.
Response 4: This was done as requested. In addition, we now show individual components of the pathology scores in the new supplementary Figure 2. This figure is now also listed in the text of the Results section:
Lines 147-148: “(…; see Supplementary Figure 2 for individual components of the pathology score).
Comment 5: In general the introduction is too short and recent reviews on the argument are missing. In the discussion authors do not clarify enough how the results obtained can impact on the research regarding bacterial infection and sepsis. Details regarding the role of platelets from the clinical point of view are missing.
Response 5:
In accordance with the suggestion of the reviewer we expanded the Introduction:
Lines 37-46: “Platelets play a crucial role in the pathophysiology of sepsis, extending beyond their traditional hemostatic function. During sepsis, platelets become activated due to the ongoing coagulation cascade, inflammatory response, and endothelial tissue damage. These activated platelets contribute to hyperinflammation, disseminated intravascular coagulation and microthrombosis, ultimately leading to multiple organ failure. Moreover, platelets can interact with innate immune cells, modulating their functions and promoting a pro-inflammatory phenotype. Additionally, platelets release proinflammatory peptides and cytokines, further influencing the immune response. The complex interplay between platelets, coagulation, and inflammation in sepsis makes them a potential target for therapeutic interventions”.
Discussion: we expanded the final part of the first paragraph (new section underlined);
Lines 180-188: “Yet, our data indicate that platelet specific Lkb1 deficiency does not affect platelet responses or host defense during K. pneumoniae infection, suggesting that alternative pathways or compensatory mechanisms may supersede or mask Lkb1's function in the complex physiological context of a live organism facing a bacterial challenge. In this context, it should be noted that platelets can be activated through a variety of agonists and mechanisms during sepsis, including adenosine diphosphate, thromboxane A2, cytokines, matrix metalloproteinases, acidosis and interactions with endothelial cells, conditions that are not or incompletely captured in “ex vivo” stimulations”.
In addition, we added a final paragraph to the Discussion:
Lines 231-234: “Previous work has suggested that therapeutic interventions that target activated platelets might reduce hyperinflammatory and procoagulant aberrations in sepsis. We here report experimental data that argue against a significant role for Lkb1 in platelet activation and host defense in pneumonia-derived sepsis”.

Round 2
Reviewer 1 Report
Comments and Suggestions for Authors
Unfortunately, authors did not respond appropriately the issues raised by reviewer.
Author Response
Reviewer 1 (who was Reviewer 2 in the first “round”) states “Unfortunately, authors did not respond appropriately the issues raised by reviewer”.
Response: We are not sure how to respond to this comment since the reviewer does not specify which issues are not addressed appropriately. Below we repeat his/her original comments and our response in blue italics (visible in accompanying word file), with our new response underneath.
Earlier Comment 1: They showed Lkb1 deficiency in platelets is associated with platelet hyperactivation in response to GPVI stimulation, even though I can’t see the figure. However, they commented that Platelet specific Lkb1 deficiency does not impact the host response during pneumonia sepsis. However, in table 1, even though there is no statistical difference, there are some trends btw two groups, e.g. IL-6 (105 vs. 594). The same trends were observed in IL-10, CCL2, and TNF. I am not sure two groups have no real difference or numbers are short.
Earlier Response 1: We now discuss this issue in a limitations paragraph in the Discussion:
Lines 225-230: “Our study has limitations. While differences between genotypes after infection were not significant, some trends were seen (for example higher median CFU counts in lungs and BALF, and higher median plasma IL-6 levels in platelet specific Lkb1-deficient mice at 40 hours); expansion of the sample size would possibly reveal significant differences between groups”.
New response: We wonder whether this comment relates to the statement that the reviewer cannot see the figure showing platelet hyperactivation in response to GPVI stimulation. If so, then something must be wrong in the submission system, since in our files this figure is visible. With regard to possible differences in cytokine and chemokine levels between groups, we would like to pint out that P values weren’t even close to significant. After consulting with a clinical epidemiologist in our group, we concluded that retrospectively it is impossible to accurately assess whether increasing the number in the groups would have led to significant differences. We have kept the section in our limitations section added in the previous round of revisions:
Lines 225-230: “Our study has limitations. While differences between genotypes after infection were not significant, some trends were seen (for example higher median CFU counts in lungs and BALF, and higher median plasma IL-6 levels in platelet specific Lkb1-deficient mice at 40 hours); expansion of the sample size would possibly reveal significant differences between groups”.
Earlier Comment 2: In clinical aspects, what is the implication of the discrepancies btw platelet hyperactivation in response to GPVI stimulation and no difference in in vivo model?
Earlier Response 2: We addressed the discrepancy between our in vitro and in vivo data in the first paragraph of the Discussion: “Yet, our data indicate that platelet specific Lkb1 deficiency does not affect platelet responses or host defense during K. pneumoniae infection, suggesting that alternative pathways or compensatory mechanisms may supersede or mask Lkb1's function in the complex physiological context of a live organism facing a bacterial challenge”. We now add to this the following:
Lines 184-188: “In this context, it should be noted that platelets can be activated through a variety of agonists and mechanisms during sepsis, including adenosine diphosphate, thromboxane A2, cytokines, matrix metalloproteinases, acidosis and interactions with endothelial cells, conditions that are not or incompletely captured in “ex vivo” stimulations”.
New response: We would be very happy to expand the discussion on this topic; for this we would respectfully request suggestions by the reviewer. We believe we have addressed the most likely explanations of the different in vitro and in vivo results.
We look forward to your reaction.
Sincerely,
Osoul Chouchane & Tom van der Poll

Reviewer 2 Report
Comments and Suggestions for Authors
The authors addressed my question.
Of note, I still not see the flow citometry example of the staining represented with box plot. This could be added if also the editor thinks it is necessary.
Author Response
Reviewer 2 (who was Reviewer 1 in the first “round”) has no further comments and is satisfied by our response. We would like to thank the reviewer for adding the the quality of our manuscript.